# Matrix Metalloproteinases in Pulmonary and Central Nervous System Tuberculosis—A Review

**DOI:** 10.3390/ijms20061350

**Published:** 2019-03-18

**Authors:** Ursula K. Rohlwink, Naomi F. Walker, Alvaro A. Ordonez, Yifan J. Li, Elizabeth W. Tucker, Paul T. Elkington, Robert J. Wilkinson, Katalin A. Wilkinson

**Affiliations:** 1Neuroscience Institute, University of Cape Town, Faculty of Health Sciences, Anzio Road, Observatory 7925, South Africa; uk.rohlwink@uct.ac.za; 2TB Centre and Department of Clinical Research, London School of Hygiene and Tropical Medicine, Keppel St, London WC1E 7HT, UK; Naomi.Walker@lshtm.ac.uk; 3Department of Pediatrics, Johns Hopkins University School of Medicine, Baltimore, MD 21287, USA; aordone2@jhmi.edu; 4Center for Tuberculosis Research, Johns Hopkins University School of Medicine, Baltimore, MD 21287, USA; etucker9@jhmi.edu; 5Faculty of Health Sciences, University of Cape Town, Anzio Road, Observatory 7925, South Africa; joshualyf17@gmail.com; 6Department of Anesthesiology and Critical Care Medicine, Johns Hopkins University School of Medicine, Baltimore, MD 21287, USA; 7Division of Pediatric Critical Care, Johns Hopkins All Children’s Hospital, St. Petersburg, FL 33701, USA; 8NIHR Biomedical Research Centre, School of Clinical and Experimental Sciences, Faculty of Medicine, University of Southampton, Southampton, SO16 6YD, UK; P.Elkington@soton.ac.uk; 9Wellcome Centre for Infectious Diseases Research in Africa, Institute of Infectious Diseases and Molecular Medicine, University of Cape Town, Cape Town 7925, South Africa; 10The Francis Crick Institute, 1 Midland Road, London NW1 1AT, UK; 11Department of Medicine, Imperial College London, London W2 1PG, UK; r.j.wilkinson@imperial.ac.uk

**Keywords:** tuberculosis, matrix metalloproteinases, tuberculous meningitis, HIV-TB-associated IRIS, extracellular matrix breakdown, adult, pediatric, lung, central nervous system

## Abstract

Tuberculosis (TB) remains the single biggest infectious cause of death globally, claiming almost two million lives and causing disease in over 10 million individuals annually. Matrix metalloproteinases (MMPs) are a family of proteolytic enzymes with various physiological roles implicated as key factors contributing to the spread of TB. They are involved in the breakdown of lung extracellular matrix and the consequent release of *Mycobacterium tuberculosis* bacilli into the airways. Evidence demonstrates that MMPs also play a role in central nervous system (CNS) tuberculosis, as they contribute to the breakdown of the blood brain barrier and are associated with poor outcome in adults with tuberculous meningitis (TBM). However, in pediatric TBM, data indicate that MMPs may play a role in both pathology and recovery of the developing brain. MMPs also have a significant role in HIV-TB-associated immune reconstitution inflammatory syndrome in the lungs and the brain, and their modulation offers potential novel therapeutic avenues. This is a review of recent research on MMPs in pulmonary and CNS TB in adults and children and in the context of co-infection with HIV. We summarize different methods of MMP investigation and discuss the translational implications of MMP inhibition to reduce immunopathology.

## 1. Introduction

Tuberculosis (TB) remains one of the top 10 causes of death globally, with 1.6 million deaths attributed to TB in 2017 [1]. Although the overall incidence of the disease is decreasing, there were over 10 million new cases of TB reported in 2017 [1], suggesting that more work needs to be done on disease prevention and cure. Tuberculous meningitis (TBM) is the most severe form of this disease and leaves many surviving adults and children with severe neurological impairment [2]. The inflammatory processes occurring during TB infection in the lungs and the neuro-inflammatory response as a result of dissemination to the brain play key roles in disease outcome. In pulmonary TB (pTB), matrix metalloproteinases (MMPs) are considered important in cavitary lung disease and the subsequent spread of *Mycobacterium tuberculosis* (*Mtb*) from the lung parenchyma into the airways from whence they can be expectorated to perpetuate infection [3]. They also contribute to the breakdown of the protective blood brain barrier (BBB) and brain tissue destruction consequent to TBM [3]. In this review, we discuss the current status of MMP research in pulmonary and central nervous system (CNS) TB, and their potential dual role in pathology as well as neurodevelopment and recovery. We also summarize potential novel treatment strategies that may target MMPs.

### MMPs

MMPs are a superfamily of zinc- and calcium-dependent proteolytic enzymes that is well conserved across species with 25 vertebrate MMPs (24 of which are found in humans) characterized to date [4]. Several MMPs, including MMP-1, -3, -7, and -8, are located within a single gene cluster, highlighting their essential role in fundamental physiologic processes [4]. Their primary function in humans is degradation of the extracellular matrix (ECM) [5], which not only contributes to health by providing tissue homeostasis but also has a role in many pathologic conditions [4]. MMPs fall under the general category of metalloproteinases (MPs) of the Metzincin family, which function both on cell surfaces (as sheddases, which release growth factors, death receptors, and death-inducing ligands), and within the ECM [6]. They are divided into five sub-families, namely collagenases (MMP-1, -8, and -13), gelatinases (MMP-2 and MMP-9), stromelysins (MMP-3 and MMP-10), elastases (MMP-7 and MMP-12) and membrane-type MMPs (MT-MMP-1 to -5) [7,8].

MMPs are synthesized and released as proenzymes, known as zymogens, that are subsequently activated when the zinc-thiol interaction between the catalytic and the pro-peptide domain is disrupted by cleavage of the pro-peptide domain [8,9,10]. Due to the destructive nature of excessive activity, MMPs are controlled by gene expression (transcriptional and post-transcriptional regulation), proenzyme activation, and innate inhibitors. They are inhibited by α2-macroglobulin in the plasma or tissue inhibitors of metalloproteinase (TIMPs) in the tissue [4]. TIMPs are endogenous protein regulators and show tissue-specific, constitutive, or inducible expression [11]. There are four members of the TIMP family (TIMPs 1-4), which bind MMPs non-covalently to inhibit activity. TIMP-3 acts on both MMPs and tumor necrosis factor (TNF)-α converting enzyme (TACE) [9,11]. However, despite preference for specific MMPs, they are able to inhibit all MMPs, binding them in a 1:1 molar stoichiometry [11].

Physiologically, MMPs have an important role in development (blastocyst implantation, embryonic development, nerve growth), reproduction (ovulation, endometrial cycling, cervical dilatation), and maintenance of homoeostasis (wound-healing, bone remodeling, angiogenesis, and nerve regeneration) [4]. However, in various pathologies, MMPs have demonstrated a role in both immuno-protection and tissue destruction. They contribute to activation of immune mediators and leukocyte migration to sites of inflammation and infection by modulating cytokine and chemokine activity [12]. They are also implicated in numerous pathologies such as cancer invasion and metastasis, hypertension, rheumatoid and osteo-arthritis, CNS diseases, and may cause direct tissue destruction at excessive concentrations [10,12]. In the lung, a number of MMPs contribute to tissue homeostasis, such as MMP-7, -16, -19, -21, -24, -25, and -28 [4]. For example, lung epithelial cells express MMP-7 that is essential in normal wound repair and plays a part in the chemotactic recruitment of neutrophils [13]. In the CNS, MMPs are normally undetectable or present at very low levels (except for MMP-2) [10,14]. They can play a beneficial role in the CNS through their involvement with tissue repair after injury [9,14] and are also found to be elevated postnatally in rat and mouse CNS, suggesting pivotal roles in neuroplasticity and CNS development [15,16] such as synaptogenesis, synaptic plasticity, and long-term potentiation [17].

## 2. Methods of Investigation

Quantification of MMPs can be an analytical challenge for several reasons. In TBM, a number of methods of MMP investigation have been used, including ELISA, zymography and reverse zymography for serum, and cerebrospinal fluid (CSF) (Table 1). ELISAs for MMPs are, however, often limited by the absence of differentiation between active and inactive or degraded forms of MMP, and multiplexed assays are often complicated by cross-reactivity between analytes due to the protein’s common domains [8]. Additionally, the method of blood collection can impact the concentration of MMPs, with higher concentrations reported in serum compared to plasma (both heparin and EDTA) [8] due to release from leukocytes and platelets during clot activation. For MMP-1, -8, and -9, plasma is recommended [18]. Brain tissue or granuloma samples have been examined using immunohistochemistry [19,20] and MMP gene expression using PCR [20,21]. Newer techniques such as analysis of gene polymorphisms, near-infrared optical imaging, and HPLC (high performance liquid chromatography) together with inductively-coupled plasma mass spectrometry (ICPMS) are evolving and promising analytic techniques that could improve the ability to profile all MMPs [8]. In addition to human data, in vitro cell and animal models have also offered valuable insights.

## 3. MMPs in pTB

Destructive pulmonary pathology is the hallmark of human tuberculosis, and *Mtb* is relatively unique amongst pathogens for its ability to drive progressive destructive pulmonary pathology, including cavitation, in immunocompetent adults. Cavities are critical in *Mtb* pathogenesis, as they drive transmission (the most highly infectious patients are those with cavities), are immune-privileged sites with high bacterial burden, and are poorly penetrated by anti-mycobacterial drugs, leading to higher risk of treatment failure and relapse [54]. MMPs have been implicated in the pathology of pulmonary tissue damage and cavitation, as collectively, they can degrade all fibrillary components of the ECM (Figure 1). Previous reviews of MMPs in TB and mechanisms of cavitation have presented substantial evidence of MMP activity in pTB disease [3,55], summarized below:Elevated MMP concentrations (including MMP-1, -2, -8, -9) are consistently reported in respiratory fluids [sputum, broncho-alveolar lavage (BAL), and pleural fluid] from TB patients compared to patients with respiratory symptoms and/or healthy controls. Increased MMPs are associated with various markers of pTB disease severity, most significantly MMP-1 with sputum smear status, radiographic disease extent, and cavitation number in human pTB [28].Significantly increased MMP gene expression is found in human respiratory cells (alveolar macrophages, bronchial epithelial cells, and fibroblasts) and macrophages in response to *Mtb* infection and/or stimulation by conditioned media from *Mtb*-infected monocytes (CoMtb), resulting in increased MMP secretion [19,22,56]. Specifically, intracellular signaling involving p38 and extracellular signal-regulated kinase (ERK) mitogen-activated protein kinase pathway (MAPK) are important for MMP upregulation in macrophages in response to *Mtb* infection.Genetic associations implicate MMPs in TB disease risk (2G/2G MMP-1 genotype), endobronchial TB and tracheobronchial stenosis (1G MMP-1 allele), TB dissemination (MMP-9 1562C/C genotype), and post-TB chronic lung fibrosis (MMP-1 G-1607GG polymorphism).Whilst some animal models of tuberculosis have failed to fully replicate the spectrum of human pTB disease, following *Mtb* infection in human MMP-1-expressing transgenic mice, pathology was found to be more similar to pTB in humans with increased alveolar tissue damage and collagen destruction compared to wild type mice [27].

As MMPs have different cellular sources and substrate specificity, it is unsurprising that specific MMPs have been associated with different roles within the same disease. For example, MMP-1 has been most strongly implicated in the development of pulmonary cavities, whereas MMP-9 has been implicated in *Mtb* dissemination.

Here, we review in detail more recent (since 2013) studies of pTB that have further delineated cellular sources of MMPs in TB, mechanisms by which MMP activity is regulated in TB, and the importance of the cellular and extracellular environment. We also describe studies that have evaluated MMPs and matrix degradation products as TB biomarkers (Table 1, Section A).

### 3.1. Collagen Degradation is an Early Pathological Event Promoted by Cell-Matrix Adhesion in TB

Fibrillar type I, III, and IV collagen are major structural components of the human lung. In order to develop cavities, these collagen fibers must be cleaved. MMP-1 (also known as interstitial collagenase, collagenase-1) has emerged as the key MMP causing collagen degradation in pTB. In a recent study, of all the MMPs evaluated, MMP-1 was the most significantly increased in induced sputum of TB patients compared to respiratory symptomatic patients and healthy controls, along with MMP-3, which activates MMP-1 [57]. Further, MMP-1 positively correlated with chest radiograph inflammation score in HIV-uninfected patients and with sputum acid fast bacilli score and cavity frequency in HIV-infected and uninfected patients [57]. Procollagen III N-terminal propeptide (PIIINP), a matrix degradation product released during Type III collagen turnover, was increased in sputum of TB patients compared to controls and correlated with sputum MMP-1 but not with other sputum MMPs or cytokines measured, suggesting that MMP-1 is the key collagenase degrading Type III collagen in the lung [30].

Kubler et al. [32] investigated cavitary formation in a novel rabbit model of pTB, demonstrating increased MMP -1, -3, -7, -12, and -13 transcription in abnormal lung tissue. MMP-1, followed by MMP-3, was the most highly upregulated MMP with increased expression in the cavity walls compared to granulomatous tissue, and MMP-1 was found to be highly abundant in cavitary areas, supporting human data implicating MMP-1 in cavity formation. In contrast, TIMP-3 expression was lower in the cavitary compared to granulomatous areas of lung.

Al Shammari et al. [58] examined lung biopsy specimens from pTB patients and demonstrated that collagen and elastin absence mapped to areas of caseous necrosis. In transgenic mice expressing human MMP-1 on the scavenger receptor A promoter/enhancer, infection with a pathological strain of *Mtb* (recently isolated from a patient) resulted in MMP-1 expression in the lung and development of granuloma with central areas of tissue destruction containing amorphous cellular debris, consistent with caseous necrosis. Wild-type or MMP-9-expressing mice infected with the same strain of *Mtb* did not develop this pathology despite a similar extent of pulmonary inflammation, mycobacterial burden, and cytokine profiles. The primary pathological difference was loss of collagen within the MMP-1-expressing mouse granulomas. In a 3D extracellular matrix *Mtb* infection model, the addition of collagen, but not gelatin, improved monocyte survival after *Mtb* infection, suggesting that loss of collagen may promote cell death in TB. Together, this work supports the hypothesis that MMP-1 activity leads to collagen degradation as a prelude to caseous necrosis in TB rather than as a consequence, and that the integrity of the ECM may be an important determinant of the host immune response to *Mtb* [58].

Following up on this, Brilha et al. [59] found that the ECM regulated MMP activity in TB. Culture filtrate from *Mtb*-infected monocytes (CoMtb) increased monocyte adhesion to the extracellular matrix. Adhesion to Type I collagen increased MMP-1 secretion from *Mtb*-infected monocytes by 60%, MMP-7 by 57%, and MMP-10 secretion by 90%. Similarly, adhesion to fibronectin (but not Type IV collagen) increased MMP-1 secretion by 63% and MMP-10 secretion by 55%. Type I collagen adhesion increased *Mtb*-driven TIMP-1, but TIMP-2 decreased. Monocyte migration, adhesion, and type I collagen degradation were dependent on αVβ3-integrin activation, and integrin αVβ3 cell surface expression increased in *Mtb*-infected monocytes following adhesion to type I collagen and fibronectin. This suggests that *Mtb* infection promotes immune cellular interaction with the ECM propagating ECM destruction. Whilst plasma concentrations of TIMP-1 are elevated in pTB in adults and children [60,61], the concentration of antiproteases at the site of disease is evidently insufficient to prevent matrix destruction, and TIMP-1 concentrations in pulmonary secretions are reduced in TB [27]. In cell culture experiments, the increase in MMP secretion is not countered by an increase in TIMP-1 secretion [23], consistent with *Mtb* skewing the protease-antiprotease balance towards matrix breakdown.

In addition to surface-bound collagenases, MT-MMPs may drive collagen degradation in TB. MT-MMP-1 (also known as MMP-14) is expressed in TB granulomas [39]. Increased MT-MMP-1 transcript abundance was found in the sputum of pTB patients compared to controls. *Mtb* infection increased monocyte cell surface MT-MMP-1 expression and resulted in collagen degradation, which was largely MT-MMP-1-dependent. CoMtb similarly upregulated MT-MMP-1, which was p38 MAPK dependent. MT-MMP-1 expression increased cell migration in a cellular model of TB infection, therefore MT-MMP-1 may contribute to both collagenase activity and cell migration in TB [39].

### 3.2. Neutrophil-Derived MMP-8 Drives TB Immunopathology

Neutrophils are abundant in the lungs in pTB disease and can be both helpful and harmful in TB [62,63,64]. Neutrophil-derived MMP-8 (neutrophil collagenase) has now emerged as the second main collagenase contributing to TB immunopathology. A systematic investigation of the role of MMP-8 in TB [38] demonstrated that neutrophils secrete MMP-8 in response to *Mtb* infection via NF-κβ and in response to CoMtb, resulting in collagen degradation. Furthermore, elevated MMP-8 found in sputum from pTB patients correlated well with neutrophil activation markers and TB severity score and was associated with cavitation and involved with collagen breakdown compared to control sputum. Neutrophils containing MMP-8 were found at the inner wall of pulmonary cavities in TB patients and at the center of cavities in areas of caseous necrosis. Pro-catabolic AMP-activated protein kinase (AMPK) phosphorylation was identified as a key regulator of neutrophil-derived MMP-8, but there was no evidence of involvement of Akt/phosphoinositide-3-kinase (PI3K) pathway and mTOR/p70S6 kinase [38]. A further study found elevated sputum and plasma MMP-8 in patients with and without HIV infection, and two studies have implicated MMP-8 in pulmonary immunopathology associated with TB-associated immune reconstitution inflammatory syndrome (TB-IRIS—see below) [57,65]. In HIV-uninfected patients, elevated plasma MMP-8 was found in TB patients compared to respiratory symptomatic patients and healthy controls and was found to differ by gender in TB patients, with elevated MMP-8 in men compared to women [35]. Although men had a longer duration of cough than women, the association with gender was independent of cough duration. Taken with the MMP-1 data, these findings suggest that multiple collagenases contribute to lung matrix destruction in TB, potentially with macrophage-derived MMP-1 causing initial matrix breakdown and then neutrophil-derived MMP-8 augmenting lung tissue damage once early cavitation occurs.

### 3.3. Hypoxia Drives Collagenase Activity in pTB

TB granulomas in several animal models have been shown to be hypoxic [66], but the interplay with matrix turnover in human TB has only recently been investigated. The MMP-1 promoter sequence has numerous putative hypoxia response elements, which are hypoxia-inducible factor (HIF)-1-binding sites as well as NF-κB and activated protein-1 (AP-1)-binding sites [67]. HIF-1α is a heterodimeric transcription factor, which is a key regulator of the host response to oxygen deprivation. Positron emission tomography–computed tomography (PET-CT) study of human pTB lesions with the hypoxia-specific tracer [^18^F] fluoromisonidazole ([^18^F]FMISO) has demonstrated severe hypoxia in radiologically abnormal areas [67]. Experimental hypoxia or HIF-1α stabilization using dimethyloxalyl glycine (DMOG) increased expression and secretion of MMP-1 in *Mtb*-infected macrophages without affecting *Mtb* growth. A similar effect of experimental hypoxia on MMP-1 secretion in response to *Mtb*-stimulation (via CoMtb) on normal human bronchial epithelial cells was observed. *Mtb* increases HIF-1α accumulation, even in normoxia, and HIF-1α is found in TB granuloma co-localizing with epithelioid macrophages and multinucleate giant cells [67], demonstrating a complex interplay between *Mtb* infection, hypoxia, and matrix breakdown, which together can be predicted to cause lung cavitation.

Ong et al. [68] recently demonstrated that hypoxia (or HIF-1α stabilization) exacerbates neutrophil-derived *Mtb*-driven MMP-8 secretion. The formation of neutrophil extracellular traps (NETs) in response to direct infection of neutrophils by *Mtb* was decreased in the presence of hypoxia, but neutrophil phagocytosis of *Mtb* was not affected. Furthermore, hypoxia increased neutrophil survival following *Mtb* infection compared to normoxia and increased neutrophil-driven matrix destruction, suggesting that hypoxia in pTB lesions may facilitate *Mtb* survival and promote tissue destruction.

### 3.4. MMP and Cytokine Networks Enhance Tissue Damage in TB

Cellular networks lead to a tissue degrading phenotype in pTB, upregulating MMPs by a number of pathways without a compensatory increase in TIMPs. Cell-cell networks involving respiratory epithelial cells and fibroblasts in addition to immune cells result in synergistic MMP upregulation in TB [3].

MMP-10 (stromelysin-2) was not formerly well studied in TB. However, as a key activator of MMP-1, recent work demonstrated that MMP-10 drives collagenase activity in cell culture, and its upregulation in macrophages was induced by virulent *Mtb* in an ESAT-6-dependent manner via p38 and ERK MAPK [40]. Elevated MMP-10 was found in respiratory fluids (induced sputum and BAL fluid) in TB patients compared to respiratory symptomatic controls, suggesting it may play an important role in human disease operating within a proteolytic cascade.

A recent study by Fox et al. [41] identified a novel role for platelets in amplifying *Mtb*-driven MMP-1, -3, -7, and MMP-10, and interleukin (IL)-1β secretion from monocytes without increasing TIMP-1 and -2 levels. Similar to MMP-10, MMP-3 can activate MMP-1, and thus a predicted outcome was increased collagenase activity. When *Mtb* was co-cultured with monocytes and platelets in a fluorescence model using dye-quenched (DQ) collagen, significantly increased type 1 collagen degradation was found compared to incubation with monocytes or platelets alone. Platelet-derived mediators (P-selectin, RANTES, platelet-derived growth factor (PDGF)-BB) were present in BAL fluid from TB patients at increased concentrations compared to BAL from non-TB respiratory symptomatic patients and correlated with multiple MMPs (including MMP-1, -8, -9) and IL-1β.

IL-17 is expressed in human TB granulomas [42]. Supplementation with IL-17 enhanced CoMtb-induced MMP-3 secretion from human distal small airway epithelial cells without compensatory TIMP-1/2 upregulation. MMP-9 secretion was inhibited by IL-17 while MMP-1, -7, and -8 were unchanged. IL-17 enhanced CoMtb-driven MMP-3 upregulation in normal human bronchial epithelial cells via p38 MAPK-dependent and the PI3K pathway. In fibroblasts, CoMtb-driven MMP-3 secretion was augmented by both IL-17 and IL-22 [42].

In pleural TB, MMP-1, MMP-9, and TNF-α concentrations were elevated in pleural fluid compared to pleural fluid from heart failure controls [34]. MMP-1 and -9 were secreted by pleural mesothelial cells in response to *Mtb*-stimulation. TNF-α strongly correlated with MMP-1 secretion and was found to be a potent stimulus of MMP-1 upregulation by pleural mesothelial cells in response to *Mtb*. In pericardial TB, concentrations of MMP-1, MMP-2, and TIMP-1 were significantly elevated in pericardial fluid, while MMP-7 and MMP-9 were higher in blood of patients with and without HIV-infection [69]. Differential abundance was also detected at mRNA level for MMP-2, MMP-9, TIMP-1, and TIMP-2.

In summary, *Mtb* promotes MMP activity and collagen degradation via cellular networks, likely involving monocytes, macrophages, neutrophils, platelets, and epithelial cells in the airways and mesothelial cells in the pleural space. In addition to previously described roles for monocyte-derived Oncostatin M and TNF-α enhancing MMP-1 secretion from fibroblasts [70], *Mtb*-driven (ESAT-6-dependent) MMP-10 and IL-17-augmented MMP-3 may propagate collagenase activity in cellular networks via MMP-1 activation. An outstanding question is the relative contribution of direct cellular infection versus intercellular networks in causing lung cavitation in vivo.

Furthermore, whilst macrophages and stromal cells are established sources of MMPs in TB, the role of T cells in MMP-related pathology is unclear. T cells are critical to an efficacious host immune response to *Mtb*, yet they also contribute to immune-mediated tissue damage. To exemplify, a greater purified protein derivative (PPD) response associates with subsequent development of pulmonary TB [71] and immune checkpoint inhibition for cancer associates with development of active TB [72,73]. In contrast, in advanced HIV-induced immunocompromise, pulmonary cavitation rarely occurs—despite high mycobacterial load within the lung—unless immune reconstitution takes place with therapy. The knowledge gap defining what comprises a protective T cell response versus what contributes to MMP-mediated tissue damage is a challenge to the development of rational strategies that may limit pathology after treatment initiation or alternatively prevent cavitation to break the cycle of transmission.

### 3.5. Intracellular Regulation of MMP Activity in pTB

Regulation of MMP-1 along with MMP-3 and MMP-9 by the PI3K pathway in TB has been investigated in normal human bronchial epithelial cells stimulated by intercellular networks [74]. Inhibition of the PI3K p110α subunit led to increased MMP-1, whereas downstream inhibition of Akt, mTOR, and p70S6 kinase led to reduced MMP-1. MMP-3 was inhibited by blockade at all three points in the pathway, whereas MMP-9 was increased by blockade at PI3K p110α subunit, decreased by blockade at Akt, and increased by blockade at mTOR or p70S6 kinase, indicating that regulation was complex and possibly involved epigenetic mechanisms [74].

Subsequent work has demonstrated epigenetic control of *Mtb*-driven MMP-1 and -3 expression by changes in histone acetylation in macrophages and normal human bronchial epithelial cells in a cell-type specific manner [75]. *Mtb* also inhibits negative regulatory pathways in directly infected human macrophages that function to limit MMP-1 secretion by inducing microRNAs that target PI3K, mTOR-containing complex 1 (mTORC1), and MAPK-interacting kinase (MNK) pathways, which together inhibit MMP-1 gene expression [76]. Therefore, *Mtb* not only increases collagenase activity by inducing macrophage MMP-1 and -10 secretion via p38 and ERK MAPK, it also effectively switches off inhibition of MMP-1 gene expression by interfering with PI3K and MNK pathway- regulation of MMP-1 inhibition [76].

Andrade et al. [37] observed elevated plasma MMP-1 and heme oxygenase (HO)-1 in TB patients compared to latently TB-infected asymptomatic controls. There was an inverse relationship between plasma MMP-1 and HO-1, and active TB patients were found to be either plasma MMP-1-high, plasma HO-1 low, or plasma MMP-1-low, HO-1-high. HO-1 is an antioxidant that catalyses the first step of the oxidative reaction that degrades the heme group contained in several proteins (including haemoglobin and myoglobin) and is immunomodulatory [77]. In vitro, both MMP-1 and HO-1 were induced in a dose-dependent fashion by *Mtb*. HO-1 was determined to be ESAT-6-dependent and required replicating bacilli, whereas MMP-1 secretion did not. In murine macrophages, HO-1-induced carbon monoxide (CO) production inhibited MMP-1 at the transcriptional level by inhibiting activation of c-JUN/AP-1, suggesting that *Mtb*-driven HO-1 may serve to limit MMP-1 [37]. Together, these findings demonstrate the complexity of MMP regulation with a balance between stimulatory and inhibitory pathways.

### 3.6. MMPs as Biomarkers of TB Disease

MMPs have been evaluated as diagnostic biomarkers of active TB. The previously described study by Sathyamoorthy et al. [35] reported that MMP-8 was the most discriminatory MMP for active TB in plasma, reporting an area under the curve (AUC) of 0.77 (0.80 for men and 0.72 for women) in comparison to non-TB respiratory symptomatic patients by receiver operating characteristics (ROC) curve analysis.

Following the demonstration that plasma PIIINP correlated with radiographic inflammation score and sputum MMP-1 in pTB, PIIINP was explored alongside other matrix degradation products as a peripheral biomarker of pTB disease [30]. Plasma PIIINP was combined with procollagen III C-terminal propeptide (PIIICP), MMP-2, MMP-7, MMP-8, and body mass index (BMI) in a TB prediction model in comparison to a mixed group of non-TB respiratory symptomatics and healthy controls. The final model, which included only those variables predictive after adjustment for all the others, included only MMP-8 and PIIINP, giving an AUC of 0.82 (95% confidence interval CI 0.742–0.922, *p* < 0.001) by ROC curve analysis.

In the aforementioned study by Andrade et al. [37], elevated plasma MMP-1 and HO-1 were highly discriminatory for active TB compared to latent TB. However, the comparator group of latent TB patients was asymptomatic. In an analysis including patients with other granulomatous pulmonary conditions, MMP-1 was significantly elevated in active TB (*n* = 18) compared to patients with sarcoidosis (*n* = 48) but not compared with patients with non-TB mycobacterial infection (*n* = 11). TIMP-1 has been evaluated in active TB compared to patients with pneumonia. However, the sensitivity was only 62.3% for a specificity of 45.95% [60]. It is possible that a panel of biomarkers that combine elements of tissue remodeling (MMPs, TIMPs, and matrix degradation products) may be more helpful than focus on a single component given the complexity of their interactions.

A limited number of studies to date have evaluated MMPs as biomarkers of treatment response in pTB. Ugarte-Gil et al. examined sputum MMPs longitudinally in relation to culture conversion and TBScore (which is associated with mortality risk) in a study of Peruvian patients [31]. Elevated sputum MMP-2, -8, -9, and TIMP-2 at diagnosis (but not TBScore) and elevated sputum MMP-3, MMP-8, and TIMP-1 at two weeks were associated with two-week sputum culture positivity. A study of Taiwanese patients found a modest association between elevated plasma MMP-8 and sputum culture positivity, both tested at two months [36]. An association between elevated serum MMP-1 and -8 over the first six weeks of treatment and delayed sputum culture conversion has been reported [78]. Further studies to evaluate MMPs and matrix degradation products as biomarkers of TB disease are in progress.

### 3.7. MMPs in TB-IRIS

TB is the leading cause of death among HIV-1-infected persons, and the World Health Organization (WHO) African region accounted for 72% of the global burden of HIV-associated TB in 2017 [1]. There were 464,633 cases of HIV-TB notified in 2017 globally, and 84% were on antiretroviral therapy (ART), which is an essential, life-saving intervention for HIV infection. However, ART-mediated immune system recovery in the setting of *Mtb* infection may lead to a pro-inflammatory state in the form of IRIS, presenting as TB-associated IRIS (TB-IRIS), which frequently complicates the otherwise beneficial dual therapy for HIV-1 and TB [79]. Two forms of TB-IRIS are recognized: paradoxical, which occurs in patients established on antituberculosis therapy before ART, but who develop acute, recurrent, or new TB symptoms and pathology after ART initiation; and unmasking TB-IRIS in patients not receiving treatment for TB when ART is started but who present with active TB within three months of starting ART [80]. The most common and most frequently studied form of HIV-associated IRIS is the paradoxical TB-IRIS with incidence in adults ranging between (19–57%) and lower incidence in children [81]. A recent randomized controlled trial demonstrated that prednisone therapy during the first four weeks after the initiation of ART for HIV infection resulted in a lower incidence of TB-IRIS than placebo without evidence of an increased risk of severe infections or cancers [82]. The pathophysiology of paradoxical TB-IRIS remains incompletely defined. The majority of studies evaluating MMPs have been conducted in adults, and this section of the review focuses on studies related to adult TB-IRIS.

The hypothesis that dysregulated MMP activity may play a role in immunopathology and tissue destruction in TB-IRIS patients was investigated for the first time by Tadokera et al. [83] through a comprehensive analysis of MMP-1, -2, -3, -7, -8, -9, -10, -11, -12, and -13 as well as TIMP-1 and -2 gene expression in *Mtb* stimulated peripheral blood mononuclear cells (PBMC) from 22 patients who developed paradoxical TB-IRIS compared to 22 similar HIV-TB co-infected, non-IRIS control patients. Protein secretion in the form of MMP pro-enzyme content was also investigated using multiplex assays in peripheral blood mononuclear cell (PBMC) culture supernatants and serum samples using the Luminex platform and ELISA for TIMP-1 and -2. Transcript levels were increased in TB-IRIS patients for MMP-3, -7, and -10, while protein secretion was higher for MMP-1, -7, -8, and -10 in stimulated cultures compared to the control patients. Serum MMP-7 concentration was elevated in TB-IRIS, and two weeks of clinically beneficial corticosteroid therapy [84] decreased this level, although not significantly.

The association between the development of paradoxical TB-IRIS and increasing MMP-1, -2, -3, -8, and -9 concentrations during ART was investigated in plasma from a cohort of 148 HIV-infected adults with advanced HIV and TB before ART initiation and four weeks later [65]. This study found that median concentrations of MMP-1, -2, and -3 decreased, while MMP-8 and -9 increased significantly during ART, and a greater early MMP-8 increase was associated with both TB-IRIS development and decreased long term lung function following TB cure. The results were broadly consistent with the Tadokera study [83], although in that study, MMP-8 transcript abundance was not increased in IRIS in *Mtb*-stimulated PBMC. This is possibly because MMP-8 is predominantly secreted by neutrophils and present in pre-synthesized granules, thus these changes would not be identified by RNA analysis of PBMC [38].

A large systematic study of HIV-associated TB that comprised of (i) a cross-sectional study of HIV-1-infected and uninfected TB patients and controls, and (ii) a prospective cohort study of HIV-1-infected TB patients at risk of TB-IRIS evaluated plasma and sputum MMP concentrations using Luminex multiplex assays [57]. The results were analyzed in conjunction with plasma PIIINP (determined by ELISA), a degradation product of collagen type III known to be released by matrix destruction in pTB [30]. MMP activity differed between HIV-1-infected and -uninfected TB patients. In HIV-uninfected TB patients, MMP activity was prominent in the pulmonary compartment and MMP-1 was dominant, whereas HIV-1-infected TB patients had reduced pulmonary MMP-1, -2, and -9 concentrations and reduced cavitation but increased plasma PIIINP compared to HIV-1-uninfected TB patients. TB-IRIS was associated with elevated extrapulmonary extracellular MMP concentrations (measured in plasma) both before and during TB-IRIS onset. Plasma MMP-8 was the most significantly increased and correlated with plasma PIIINP concentration as well as neutrophil count. This was likely driven by *Mtb* antigen, as TB-IRIS patients with a positive urinary LAM (lipoarabinomannan) result had higher plasma MMP-3, -7, and -8 concentrations compared to urine LAM negative TB-IRIS patients. Consequently, the compartmentalization of MMP activity differs between immunocompetent adults, where it is primarily within the lung, whereas in advanced HIV infection, MMP dysregulation is more systemic. In an in vitro culture model, *Mtb*-induced matrix degradation was suppressed by the MMP inhibitor doxycycline, suggesting MMP inhibition as a possible novel host-directed therapeutic strategy for preventing and/or treating TB-IRIS [57].

Overall, these studies indicate that TB-IRIS is associated with a distinct pattern of MMP gene and protein activation. In addition to the beneficial effect of corticosteroid therapy [82], modulation of dysregulated MMP activity may represent a novel therapeutic approach to alleviate TB-IRIS in HIV-TB patients undergoing treatment.

## 4. MMPs in TBM

### 4.1. The Blood Brain Barrier (BBB)

The BBB is a highly-selective permeability barrier comprised of specialized capillary endothelial cells with tight junctions, astrocytic foot processes, and pericytes [85]. It is integral in maintaining homeostasis in the CNS by the stringent regulation of molecular access to prevent entry of toxins and organisms into the brain. Increased permeability of the BBB is considered a hallmark of neuroinflammation and CNS infection [85], and research has implicated MMPs in BBB compromise [86,87]. Astrocytes, which form an integral component of the BBB, are known to release MMPs in response to disease and may therefore contribute to local damage [12]. Research into BBB disruption in TBM is sparse, but evidence indicates that MMP levels are increased in TBM [19,43,44,46,47], suggesting that they play a key role in BBB-breakdown, thereby contributing to the pathogenesis of TBM. The level and presence of specific MMPs and their inhibitors in the cerebrospinal fluid (CSF) and blood of patients may therefore be markers of BBB compromise. However, human studies have not shown association between MMP concentrations and the CSF-serum albumin index, an accepted index of BBB breakdown [47,52,53].

### 4.2. MMPs in Neuro-Inflammation

As summarized in Figure 2, MMPs contribute to ECM destruction in CNSTB. The two MMPs that have been the focus of human TBM research to date include MMP-2 and MMP-9 and their respective inhibitors, TIMP-2 and TIMP-1. MMP-2, also known as gelatinase-A, is a constitutively-expressed molecule that is present in large quantities in the normal brain and is found in astrocytes and CSF [9]. MMP-9, or gelatinase-B, is normally absent or present at low levels in the brain. It is found to be up-regulated during neuroinflammation [6], specifically in conditions such as viral and bacterial meningitis [88,89]. Together, these gelatinases digest tenascins, proteoglycans, fibronectin, type IV collagen, and laminin found in basement membranes of the BBB and ECM of the CNS, and thus are likely to contribute to the degradation of the BBB and tissue destruction [6,19,46]. This BBB degradation drives cerebral vasogenic oedema and facilitates further influx of blood-derived inflammatory cells into the CNS. The resultant chronic fibrosis and adhesions (due to exudate formation) contribute to the complications and neurological sequelae of TBM discussed below. In contrast to MMPs, TIMPs are widely expressed in many mammalian tissues. In the CNS, TIMP-1 expression is restricted to regions of persistent neuronal plasticity, such as the hippocampus, cerebellum, and olfactory bulb [11].

Cell culture models suggest that astrocyte and microglial secretion of MMPs does not occur with direct *Mtb* infection of these cells but rather in response to *Mtb*-infected monocytes as part of a leukocyte-astrocyte and leukocyte-microglia interaction, leading to the release of MMP-9, -1, and -3, respectively [12,20,21]. This network-dependent MMP expression is driven by pro-inflammatory mediators including TNF-α and IL-1β and IL-1β’s synergistic interaction with IFN-γ in the case of MMP-9. The transcription of MMP-9 is regulated by MAPK and NF-κβ [12], while MMP-1 and -3 transcriptional regulation occurs through p38 MAPK pathway, NF-κβ and AP-1 [20,90]. In contrast to other MMPs, MMP-2 secretion from microglia is inhibited by cytokines, such as TNF-α, through p38, caspase 8, and NF-κβ signaling [91]. Additionally, TIMP expression is not influenced by these cellular networks, suggesting a possible shift to matrix degradation in response to TB. MMP-3 may also be released by apoptotic neurons, resulting in the activation of microglia [12] and contributing to BBB breakdown by digesting the basal lamina and tight-junction proteins, specifically claudin-5, occludin, and laminin-α1 [92]. The breakdown of the BBB facilitates an influx of neutrophils that exacerbate BBB damage by secreting MMP-9 (also activated by MMP-3) through a monocyte-neutrophil network regulated by MAPK and Akt-PI3K [21,92]. MMP-9 is also reported to specifically target myelin basic protein in the white matter, which can negatively affect neuronal myelination [93]. The source of MMP secretion in the CNS is therefore likely a combination of infiltrating monocytes and neutrophils as well as resident astrocytes and microglia.

### 4.3. MMPs in TBM

The role of MMPs in TBM is summarized in Table 1, section B. Elevated MMP-9 concentrations have been found in the CSF of TBM patients with the degree of elevation correlated with the severity of disease [50], neurological compromise [19,44], and brain tissue injury [43,44]. Matsuura et al. demonstrated that concentrations of MMP-2 and -9 as well as TIMP-1 (but not TIMP-2) in the CSF were increased in patients with subacute (involving tubercle bacilli or mycotic organisms) meningitis compared with controls, and MMP-9 levels associated with CNS complications (such as disturbances of consciousness and psychiatric symptoms) [43]. Similarly, a study by Lee et al. demonstrated that increased MMP-2 and -9 levels in the CSF persisted during the late course of TBM and were associated with the development of complications [46]. Price et al. demonstrated that the imbalance of MMP-9 versus TIMP-1 in the CSF of TBM patients correlates with morbidity and mortality [94]. Although Green et al. found that adjunctive dexamethasone was associated with a significant decrease in CSF MMP-9 concentrations, they did not find an association between a decline in MMP-9 concentrations and improved outcome [47].

However, the association of elevated MMP-2 and -9 concentrations with poor outcome is not consistent across TBM studies, and numerous studies have demonstrated no association between MMP concentrations and outcome [45,48,52,53]. In fact, Mailankody et al. demonstrated an association between elevated MMPs and a favorable outcome, and the authors postulated that the MMP contribution to a leaky BBB may be favorable to anti-TB drug penetration and could therefore offer an advantage to recovery [52]. In a pediatric TBM study, the beneficial effect of MMPs was corroborated by a 2.2 times greater likelihood of a better outcome associated with an increase in MMP-9 concentrations in the early weeks after hospitalization [53]. It is known that MMP-9 can mediate both protective and pathological immunity at different concentrations and in different age groups [94], and this raises questions about the role of MMP-9 in pathology—and potentially recovery—at different time points during the disease course. Similar challenges have been encountered in therapeutically targeting MMPs in oncology, reflecting the complexity of protease networks in human disease [95].

TBM is often associated with localized TB granulomas (tuberculomas) [96], which demonstrate elevated MMP-1, -3, and -9 expression [19,20,21]. In brain biopsies from patients with CNS TB, MMP-1 secretion was highest in the granuloma, and elevated concentrations of MMP-1 and -3 were associated with microglia in the peri-granuloma region and surrounding brain parenchyma and p38 positive microglia infiltrating necrotizing TB granulomas [20,91]. Using similar samples, Ong et al. also demonstrated MMP-9 secreting neutrophil presence in these CNS TB granulomas [21].

### 4.4. MMPs in TBM-IRIS

TBM-IRIS is a frequent, severe complication of ART in HIV-associated TBM patients. A prospective study of HIV-infected ART-naive patients with TBM showed that 47% (16/34) of TBM patients developed TBM-IRIS [97]. Patients who went on to develop TBM-IRIS had higher CSF neutrophil counts compared with non-TBM-IRIS patients and were more likely to have *Mtb* cultured from CSF at TBM presentation. The relative risk of developing TBM-IRIS was 9.3 (CI: 1.4–62.2) if the CSF was *Mtb* culture positive, indicating that the condition is driven by *Mtb* antigen load. This study also showed that a combination of IFN-γ and TNF-α concentrations measured in the CSF may predict TBM-IRIS, potentially paving the way for CSF immunodiagnostics. Subsequently, the concentration of 40 immune mediators in CSF (33 paired with blood) from HIV-infected patients with TBM was determined at TBM diagnosis, at initiation of antiretroviral therapy (ART, day 14), 14 days after ART initiation, at presentation of TBM-IRIS, and 14 days thereafter [49]. Among others, MMP-1, -2, -3, -7, -9, -10, -12, and -13, and TIMP-1 and -2 were analyzed using Luminex multiplex assays in CSF and plasma of patients who developed TBM-IRIS (*n* = 16) compared to those who did not (TBM-non-IRIS; *n* = 18).

Patients who developed TBM-IRIS had higher concentrations of MMP-1, -3, -7, and -10 in their blood, while MMP-9 and TIMP-1 were higher in CSF relative to blood at the time of TBM diagnosis and at the time of starting ART, as well as at the time of developing TBM-IRIS symptoms, when their concentrations increased further. However, CSF MMP-9 concentrations did not decrease following anti-TB and corticosteroid therapy and continued to rise following ART initiation, suggesting that more potent and specific therapy may be needed for the management of TBM-IRIS [98]. Overall, both at TBM diagnosis and at two weeks after ART initiation, TBM-IRIS was associated with severe compartmentalized inflammation in the CSF, elevated concentrations of cytokines, chemokines, neutrophil-associated mediators, and MMPs compared with TBM-non-IRIS, and driven by a high baseline *Mtb* antigen load [49]. A longitudinal whole-blood microarray analysis of HIV-infected patients with TBM confirmed the more abundant neutrophil-associated transcripts in patients who went on to develop TBM-IRIS from before symptom development through to TBM-IRIS symptom onset. This study also demonstrated a significantly higher abundance of transcripts associated with canonical and noncanonical inflammasomes in patients with TBM-IRIS compared to non-IRIS controls [51]. These findings, reflected at the protein and molecular level, suggest a dominant role for the innate immune system in the pathogenesis of TBM-IRIS, offer mechanistic insights into the disease, and inform potential future directions for host-directed therapy. It is notable that when patients with HIV-associated TBM were compared to those with HIV-associated cryptococcal meningitis, MMP-1 and -3 were more elevated in the former and MMP-10 in the latter, suggesting that MMP profiles may differ across meningitides of varying etiologies [99].

### 4.5. Adults and Children

Before the era of HIV, the most important risk factor for TBM was age, with young children (aged 0–4) bearing the brunt of the disease [100,101]. This is possibly due to children having an immature immune system less capable of halting TB dissemination after primary infection [100,101]. However, apart from their role in neuro-inflammation, MMPs also play a unique role in normal brain development. Several animal studies have shown that MMPs, specifically MMP-2, -3, and -9, have a crucial role in the developing brain. For example, mice lacking MMP-9 have maladapted neuronal circuitry with simplified dendritic morphology, abnormal synaptic structure and function, and enhanced excitability [102]. MMP-3 and -2 are expressed in the cortical neurons of young mice and are required for appropriate axonal outgrowth and orientation [16,103]. Van Hove et al. also found that MMP-3 deficient mice developed deficits in balance and motor performance related to abnormal cerebellar development [104]. Despite the high prevalence of TBM in children, MMPs during pediatric TBM infection have not been well studied. In our recent pediatric TBM study [53], the positive association between increasing MMP-9 concentrations and good outcome may suggest a role in recovery and ongoing neurodevelopment, possibly including angio- and myelino-genesis, the growth of axons and synaptic plasticity. Similarly, the positive correlation found between MMP-2 and TIMP-2 may also suggest a physiological role in remodeling ECM in the developing CNS [53]. However, further studies with appropriate comparison with adults are required.

## 5. Treatment

### 5.1. MMP Inhibition in pTB

Multiple host-directed therapies centered on modulating protease activity have been evaluated in animal models (Table 2) as adjuncts to antibacterial treatment in TB with the prospect of beneficial immune modulation leading to enhanced bacterial clearance and the reduction of tissue damage and inflammation, thereby potentially shortening treatment duration [105]. This is challenging given the complexity of the host immune response. For example, multiple drugs that target upstream inflammatory pathways can have an effect on MMP expression by indirectly inhibiting their activity (e.g., prednisone, doxycycline, vitamin D, rapamycin) [106]. Similarly, non-specific immunomodulation of the host response with corticosteroids has been evaluated in pTB, with most studies showing no added benefit compared to standard treatment alone [107]. Conversely, in cases of extra-pulmonary TB, such as TBM and TB pericarditis, adjuvant use of corticosteroids has been shown to reduce mortality and morbidity, leading to their common use in current clinical practice [108]. Doxycycline, a tetracycline antibiotic, has been shown to non-specifically inhibit multiple MMPs independently of its antimicrobial activity [109]. However, doxycycline also has immunomodulatory effects on other host inflammatory pathways, confounding its potential MMP-specific effects [110]. Walker et al. reported that doxycycline modulated MMP expression in *Mtb*-infected cells and reduced mycobacterial growth in vivo [28]. However, additional studies are necessary to evaluate the potential of doxycycline as MMP-inhibitor therapy in TB. Indirect inhibition of MMPs has also been reported with a phosphodiesterase-4 inhibitor (CC-3052) in a rabbit model of TB [111]. CC-3052 plus isoniazid significantly reduced the extent of immune pathology and the expression of MMPs compared with antibiotics alone.

A more specific approach to evaluating MMP-inhibition in pTB has also been attempted with multiple specific MMP inhibitors (e.g., batimastat, cipemastat, marimastat) using different animal models of pTB. However, these studies have generated conflicting data, and the differences in treatment regimens, bacterial burden, host immune response, and variables measured make direct comparisons difficult (Table 2).

Initial experiments focused on monotherapy with MMP inhibitors in mouse models of pTB. Batimastat (BB-94), a potent broad-spectrum inhibitor, showed promising results in cancer models and was one of the first MMP inhibitors to enter clinical trials [112]. However, its poor solubility and side effects associated with intraperitoneal dosing led to its discontinuation. In pTB models, Hernandez-Pando et al. used *Mtb*-infected Balb/c mice to show a trend towards increased mortality if the treatment with batimastat was started immediately after infection with *Mtb* [113]. However, the increased mortality and histological differences were not evident between treated and control groups if batimastat was initiated a month after infection. Overall, there was a shift towards a type-2 cytokine profile and delayed granuloma formation in treated mice compared to controls. Subsequent studies by Izzo et al. using batimastat in C57BL/6 mice infected with *Mtb* showed reduced bacterial burden and smaller granulomas with more collagen deposition at earlier stages (day 40), but no differences were seen at day 60 post-infection [114]. However, a follow up study by the same group did not observe a reduced bacterial burden in the same mouse model when batimastat was initiated right after *Mtb* infection [115].

Recently, Urbanowski et al. used cipemastat, an orally available MMP inhibitor, in a rabbit model of cavitary pTB with repeated aerosol exposures to *Mtb* [116]. Compared to controls, monotherapy with cipemastat during week five to ten post-infection did not reduce the rates of cavitation evaluated by CT, mitigate disease severity, or visibly affect the collagen structure of the cavitary wall (examined post-mortem). Conversely, the cipemastat-treated group had a trend toward worse cavitary disease. Ordonez et al. used cipemastat monotherapy in the C3HeB/FeJ mouse model that develops cavitary lesions after aerosol infection with *Mtb* [117]. Similar to rabbits, cipemastat-treated mice had worse cavitation and more disease severity compared to controls.

While monotherapy with MMP inhibitors suggested worse outcomes, the use of multiple MMP inhibitors (marimastat, batimastat, or SB-3CT) in combination with anti-TB therapy have had synergistic effects in reducing the *Mtb* burden in a pTB mouse model [118]. While no effect on bacterial burden was observed with marimastat treatment alone, its combination with isoniazid significantly reduced lung colony forming units compared to isoniazid alone. Marimastat also reduced vascular leakage surrounding TB granulomas and increased the concentration of isoniazid in *Mtb*-infected lungs compared to controls. Similarly, targeting MMP-9 with a specific antibody in addition to anti-TB therapy (rifampin, pyrazinamide, and isoniazid) reduced pulmonary bacterial burden and significantly lowered relapse rates in a cavitary mouse model of pTB [117]. Therefore, MMP inhibition in conjunction with antibiotic therapy may be beneficial, whilst MMP inhibition monotherapy seems to be harmful, further demonstrating the complexity of events in vivo and the importance of performing translational preclinical studies that most closely reflect the intended therapeutic use in patients.

### 5.2. MMP-Inhibition in CNS TB

Given the important role MMPs have in the degradation of the BBB and tissue destruction in CNS TB, MMPs are a possible target for adjunctive treatment. In vitro CNS TB studies have demonstrated that dexamethasone leads to a decrease in MMP-1 and -3 mRNA expression and secretion without impacting TIMP expression. This has been suggested as a mechanism by which steroids improve short-term mortality from TBM [20]. NF-κβ inhibition and anti-TNF-α have been shown to decrease neutrophil expression of MMP-9, which is postulated as one mechanism by which to decrease CNS destruction [21]. Given the pro-inflammatory network involvement in MMP-9 expression, it has been suggested that targeting this network may offer opportunities for MMP regulation without directly interfering with the role of MMPs in the host defense against *Mtb* [12]. In TBM-IRIS, protein and transcriptome studies have highlighted the role of a neutrophil driven and inflammasome-associated inflammatory response in disease pathogenesis [49,51]. ECM destruction due to neutrophil secretion of MMP-9 is suggested to contribute to brain tissue damage, and pyroptosis driven by caspase-1 secretion may cause further injury. Agents specifically targeting the inflammasome could therefore offer another potential avenue for host-directed therapy.

Although MMP secretion may be affected by several medications, there has also been interest in developing drugs to specifically target MMPs. Paul et al. evaluated the effect of the MMP inhibitor Batimastat (BB-94) on BBB breakdown (as measured by Evans Blue leakage) and increased intracranial pressure (ICP—measured via catheter) using a rat model of meningococcal meningitis. The disruption of the BBB (and subsequent pathological changes) was ameliorated in animals treated with BB-94 in a dose-dependent manner [120]. Majeed et al. used an MMP-9 specific inhibitor SB-3CT [50,121], which appeared more effective than dexamethasone in that it suppressed MMP-9 and enhanced the effect of anti-TB drugs on clearing TB bacilli [121].

## 6. Conclusions

Evidence suggests that MMPs are an important component of the inflammatory response to Mtb infection and the consequent breakdown of the ECM in both the lungs and the brain. Their potential role in brain development suggests that MMPs may be relevant in both normal physiology and pathophysiology. However, they form part of an elaborate network of inflammation and injury and constitute one element of a complex inflammatory process in TB. A better understanding of what contributes to MMP-mediated tissue damage and the role of MMPs in the broader disease context is needed to develop targeted treatment strategies that may limit pathology or prevent cavitation and transmission without the risk of unexpected effects.

## Figures and Tables

**Figure 1 ijms-20-01350-f001:**
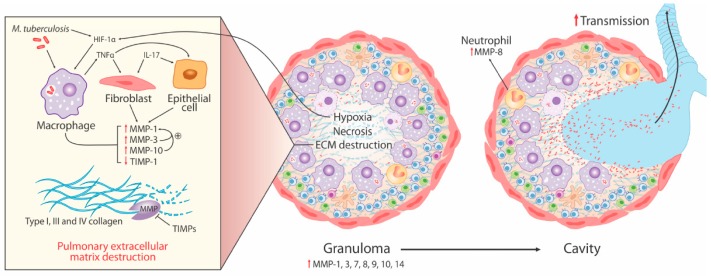
MMPs contribute to extracellular matrix (ECM) destruction in pulmonary TB (pTB). Lung TB granulomas comprise mycobacterium tuberculosis (*Mtb*) infected macrophages (purple), dendritic cells (beige), T (blue) and B (green) lymphocytes, and neutrophils (orange). These cells secrete numerous MMPs including MMP-1, -3, -7, -9, and -10, which may act in a proteolytic cascade. Evidence demonstrates their association with lung ECM breakdown, necrosis, and pTB disease severity. MMP-1 drives degradation of fibrillary type I, III, and IV collagen and is the key pulmonary collagenase. Neutrophil derived MMP-8 also contributes to collagen degradation, and both MMP-1 and -8 are involved in cavity formation, which facilitates the transmission of *Mtb* by releasing the bacilli into the airways. Hypoxia augments monocyte and neutrophil MMP secretion acting through the hypoxia inducible factor (HIF)-1α transcription factor, an important regulator of the host response to oxygen deprivation. *Mtb* also promotes MMP activity via cellular networks involving immune and stromal cells. For example, lung fibroblasts have demonstrated upregulated MMP gene expression in response to *Mtb* infection or in the presence of monocytes infected with *Mtb* and MMP-1 secretion in response to *Mtb*-induced TNF-α.

**Figure 2 ijms-20-01350-f002:**
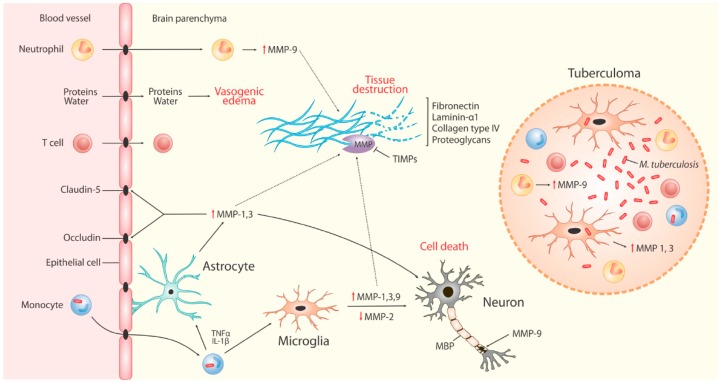
MMPs contribute to extracellular matrix (ECM) destruction in CNS TB. *Mtb* bacilli cross the protective blood brain barrier (BBB) through various mechanisms, including in infected monocytes. In the presence of these infected monocytes and driven by TNF-α and interleukin (IL)-1β, brain microglia and astrocytes respond as part of a leukocyte-microglia and leukocyte-astrocyte interaction by secreting MMP-1, -2, and -9, and MMP-1 and -3, respectively. These MMPs contribute to the breakdown of ECM proteins of the brain parenchyma, including fibronectin, laminin-α1, collagen type IV, and proteoglycans, and are pro-apoptotic to neurons (indicated in red text as “cell death”) with the resultant tissue destruction adding to the cerebral inflammatory response. MMP-9 is also known to attack myelin basic protein (MBP), an integral component of the myelin sheath insulating neurons. Further, MMPs degrade claudin-5 and occludin, thereby compromising the tight BBB and contributing to the influx of proteins and water resulting in vasogenic edema, as well as further blood-derived inflammatory cells, including neutrophils. Neutrophils secrete MMP-9 in the parenchyma as well as in TB granulomas (tuberculomas), in which *Mtb*-infected microglia have also been found to secrete MMP-1 and -3. Contrary to the other MMPs, MMP-2 expression is suppressed through cytokines like TNF-α. Up-arrow indicates increased concentrations, down-arrow indicates decreased concentrations.

**Table 1 ijms-20-01350-t001:** Summary of recent matrix metalloproteinase (MMP) studies in pulmonary tuberculosis (TB) and central nervous system (CNS) TB.

Reference	CNS or pTB	Subjects	Samples	Method of Investigation	Analytes	Key Findings
Pulmonary TB (pTB)						
Elkington et al. [22,23]	pTB	Adults (*n* = 6 proven pTB patients, 6 controls with cancer diagnosis)	Lung tissue from biopsy	Immunohistochemistry	MMP-1 MMP-7	Study examined affected lung in *Mtb* vs unaffected lung in cancer patients MMP-1 and MMP-7 present in macrophages and Langhans giant cells in granuloma, and MMP-1 in adjacent epithelial cells, in PTB cases only
Kuo et al. [24]	pTB	Adults (*n* = 101 confirmed pTB cases—38 with endobronchial TB, 68 without). All HIV negative	Blood	Genotyping	MMP-1 DNA (G-1607 GG) sequence single nucleotide polymorphisms	MMP-1 1G genotype was associated with endobronchial TB on bronchoscopy MMP-1 1G genotypes were associated with a 9.86-fold increased risk of developing tracheobronchial stenosis, in patients with endobronchial TB
Wang et al. [25]	pTB	Adult (*n* = 98 pTB cases, 49 healthy controls). All HIV negative	Blood	Genotyping	*MMP-1*(G-1067GG) single nucleotide *MMP-12*(Asn357Ser), *MMP-9*(C-1562T) polymorphisms	MMP-1 (-1607G) polymorphism increased the risk of moderate and advanced lung fibrosis at one year in pTB cases—the odds increased by 3.80 and 6.02 fold, respectively for one copy and this remained after adjustment for age, sex, initial disease score on chest radiograph, sputum bacterial load, smoking status and presence of diabetes MMP-9 and -12 polymorphisms were not associated with increased risk of developing lung fibrosis
Ganachari et al. [26]	pTB	Adults (*n* = 894 pTB cases, 1039 PPD+ controls collected from 2 sites). All HIV negative	Blood	Genotyping Immunohistochemistry of lymph node samples for MMP-1 and MCP-1	-2518A>G SNP in MCP-1 (rs1024611) -1607_1608insG variant in *MMP-1* (rs1799750), and 42 genomic control SNPs	MMP-1 allele 2G associated with TB disease MMP-1 2G/2G genotype associated with increased lymph node MMP-1 in active TB cases compared to other genotypes
Elkington et al. [27]	pTB	Adults (*n* = 33 HIV uninfected pTB cases, 32 respiratory symptomatic controls	Induced sputum and BAL	Luminex (concentrations normalized to total protein)	MMP-1 MMP-2 MMP-3 MMP-7 MMP-8 MMP-9 MMP-12 TIMP-1 TIMP-4	MMP-1, -3: pTB > controls MMP-2, -8, -9, -12, TIMP-1 and-2: pTB ≈ controls MMP-13, TIMP-3 and -4 undetectable
Walker et al. [28]	pTB	Adults (*n* = 23 pTB cases, 21 controls—mixed healthy and respiratory symptomatic). Mixed HIV status	Induced sputum	Luminex (concentrations normalized to total protein)	MMP-1 MMP-2 MMP-3 MMP-7 MMP-8 MMP-9 MMP-12 MMP-13	MMP-1, -2, -3, -8: pTB > controls irrespective of HIV status MMP-7 and -9: pTB ≈ controls MMP-12 and -13 undetectable In advanced HIV (CD4 < 200), pTB patients had relatively lower sputum MMP-1, -2, -8 and -9 MMP-1 and -2 increased in pTB patients with cavities versus no cavities, and correlated with chest x-ray inflammation score
Ganachari et al. [29]	pTB	Adults (*n* = 224 pTB cases, 42 controls). HIV negative	Blood	Genotyping	-2518A>G SNP in *MCP-1* (rs1024611) -1607_1608insG variant in *MMP-1* (rs1799750) and 42 genomic control SNPs	Greater proportion of patients with severe disease carried the two locus genoype -2518 *MCP-1* GG and -1607 *MMP-1* 2G/2G, which was also associated with delayed sputum smear conversion and increased fibrosis
Seddon et al. [30]	pTB	Adults (*n* = 78). Mixed HIV status	Induced sputum Plasma	Luminex ELISA	MMP-1 MMP-2 MMP-3 MMP-7 MMP-8 MMP-9 MMP-10 Also PINP, PIIINP, PIIICP, CTX-I, CTX-III, EMMPRIN	Plasma PIIINP correlated with radiographic inflammation score and sputum MMP-1 in pTB Plasma MMP-8 and PIIINP predictive of pTB diagnosis, AUC of 0.82 (95% confidence interval 0.742–0.922, *p* < 0.001) by ROC curve analysis
Ugarte-Gil et al. [31]	pTB	Adults (*n* = 68 HIV negative pTB cases, 69 healthy controls) Longitudinal study, follow up at 2, 8 and 24 weeks	Induced sputum	Luminex (adjusted for total protein) ELISA	MMP-1 MMP-2 MMP-3 MMP-7 MMP-8 MMP-9 TIMP-1 TIMP-2	MMP-1, -2, -3, -8, and -9: pTB cases > controls TIMP-1 and -2: pTB cases > controls Sputum MMP concentrations decreased with TB treatment but TIMP concentrations initially rose Elevated sputum MMP-2, -8, -9 and TIMP-2 at TB diagnosis, and elevated sputum MMP-3, MMP-8 and TIMP-1 at two weeks were associated with two-week sputum culture positivity
Kubler et al. [32]	pTB	Adults (*n* = 97 pTB cases, 14 latent TB and 20 healthy controls without latent TB)	Plasma	ELISA	MMP-1 MMP-3 MMP-7 MMP-8 MMP-9 TIMP-1 TIMP-2 TIMP-3 TIMP-4	MMP-1/TIMP1-4 ratios: pTB > latent TB and healthy controls
Singh et al. [33]	pTB	Adults (*n* = 17 confirmed pTB cases, 18 respiratory symptomatic controls. All HIV uninfected)	BAL Fluid	Not specified	MMP-1 MMP-2 MMP-3 MMP-7 MMP-8 MMP-9 MMP-12 MMP-13	median MMP-1, -2, -3, -7, -8, and -9: pTB cases > controls
Chen et al. [34]	Pleural TB	Adults (*n* = 18 TB pleuritis cases, 18 controls with congestive heart failure and pleural effusion)	Pleural fluid	ELISA	MMP-1 MMP-7 MMP-9 Also TNF-α	Elevated TNF-α, MMP-1 and -9, which correlated with the size of the effusion in cases MMP-7: cases ≈ controls TNF-α and MMP-1 positively correlated with degree of pleural fibrosis at 6 months in cases
Sathyamoorthy et al. * [35]	pTB	Adults (*n* = 151 pTB cases, 109 symptomatic controls and 120 healthy controls)	Plasma	Luminex	MMP-1 MMP-3 MMP-7 MMP-8 MMP-9 MMP-10 MMP-12 MMP-13	Plasma MMP-8: pTB cases > respiratory symptomatic and healthy controls MMP-1: pTB cases > healthy controls only Gender specific differences in MMPs: MMP-8 in men > women
Lee et al. [36]	pTB	Adults (*n* = 167, HIV negative, culture-confirmed, drug sensitive pTB)	Blood	Luminex	MMP-1 MMP-3 MMP-8 MMP-9 MMP-12 Also cytokines and cytotoxic mediators	26 cases were smear positive at 2 months (15.6%) RANTES concentration at diagnosis and MMP-8 concentration at 2 months predicted 2-month culture status, AUC: 0.725 (0.624–0.827), and 0.632 (0.512–0.753) respectively
Andrade et al. * [37]	pTB	Brazilian adults (*n* = 63 active PTB, 15 individuals with LTBI, 10 healthy controls) Indian adults (*n* = 97 active PTB, 39 with LTBI, 40 uninfected healthy controls) North American adults (*n* = 18 culture-confirmed TB, 11 non-tuberculous mycobacteria infection [NTM], 48 pulmonary sarcoidosis) All HIV negative	Plasma	Luminex and ELISA	MMP-1 MMP-8 MMP-9 TIMP-1 TIMP-2 TIMP-3 TIMP-4 HO-1 and others	MMP-1 (Brazilian cohort) and HO-1 (Indian cohort) were elevated but inversely related (both cohorts) in PTB patients compared to healthy controls (including those with latent TB infection) MMP-1: pTB > sarcoid but pTB ≈ NTM (in American cohort)
Ong et al. [38]	pTB	Adults (*n* = 5 pTB cases) Adults (*n* = 51 pTB cases, 57 healthy controls or a subset of 11 patients in each group for collagenase experiments). All HIV negative	Lung biopsies Induced sputum	Immunohistochemistry Luminex DQ collagen degradation assay	H&E and anti-neutrophil elastase MMP-8 MMP-9 (adjusted for total protein) Also myeloperoxidase (MPO) and neutrophil gelatinase associated lipocalin (NGAL)	MMP-8 co-localized with neutrophils at the inner surface of TB cavities Neutrophils were also immunoreactive for MMP-9 Neutrophil activation markers MPO and NGAL: pTB cases > controls and strongly correlated with sputum MMP-8 Sputum MMP-8: cavities > no cavities Induced sputum collagenase activity: pTB cases > controls Collagenase activity correlates with MMP-8 concentration and is reduced by MMP-8 neutralization
Sathyamoorthy et al. [39]	pTB	Adults (*n* = 15 pTB cases, 10 controls Adults (*n* = 5 pTB cases, 5 controls)	Induced sputum Lung biopsy	RT-PCR Immunohistochemistry	MT-MMP-1 (MMP-14)	MT-MMP-1 RNA: pTB cases > controls Granuloma cell MT-MMP-1 immunoreactivity: pTB cases > controls—only alveolar macrophages were positive
Brilha et al. [40]	pTB	Adult PTB vs control (respiratory symptomatic and healthy) South African cohort; Induced sputum, mixed serostatus as described in Walker et al. [28] Indian cohort: BAL as described in Singh et al. [33] Second South African adult cohort: Induced sputum for RNA (11 pTB patients and 17 healthy controls—all HIV negative)	Induced sputum and BAL	Luminex for MMP concentrations, RT-PCR for RNA	MMP-10	Induced sputum secreted MMP-10 and MMP-10 RNA: pTB cases > controls BAL MMP-10: pTB cases > controls
Fox et al. [41]	pTB	Peruvian cohort: Plasma from adults (*n* = 50 pTB patients 50 and matched asymptomatic PPD negative controls) Indian cohort: BAL fluid from adults (*n* = 15 pTB patients and 15 matched respiratory symptomatic controls)	Plasma and BAL Fluid	Luminex	MMP-9 and platelet-derived growth factor (PDGF)-BB, RANTES, P-selectin, platelet factor-4 (PF4), Pentraxin-3 (PTX3)	Plasma MMP-9 correlated with platelet factors, PF4, PDGF-BB and PTX3 In BAL fluid, P-selectin concentrations correlated with IL-1β, MMP-1, -3, -7, -8, and -9; PDGF-BB concentrations correlated with MMP-1, -3, -8, and -9; RANTES concentrations correlated with MMP-1, -8, and -9 as well as IL-1β
Singh et al. [42]	pTB	Adults (*n* = 5 pTB cases, 5 non-TB controls	Lung tissue	Immunohistochemistry	MMP-3 IL-17	IL-17 and MMP-3: pTB > control tissue
Tuberculous meningitis (TBM)						
Matsuura et al. [43]	CNS	Adults (*n* = 21 meningitis cases [7 TBM], 30 controls)	CSF	Gelatin zymography Immunohistochemistry (demonstrated immunoreactivity for MMP-9 and -2 for infiltrating mononuclear cells)	MMP-9 MMP-2 TIMP-1 TIMP-2	MMP-9, -2 and TIMP-1: cases > controls TIMP-2: cases ~ controls MMPs correlated positively with respective TIMPs No correlations between analytes and proteins/cell counts MMP-9 and TIMP-1 concentrations positively associated with neurological complications
Price et al. [44]	CNS	Human monocytic (THP-1) cells (in-vitro study) Adults (*n* = 23 TBM, 12 bacterial meningitis, 20 viral meningitis)	CSF	Northern Blot Western Blot Gelatin zymography ELISA (for TIMP-1)	MMP-9 TIMP-1 (MMP-2)	*In-vitro study:* MMP-9 secretion increased in TB-infected cells at 24 h (compared to controls) No difference in TIMP-1 secretion between TB-infected cells and controls at 24 h (suggesting net proteolytic activity). Moderate increase (5× compared to controls) at 48 h MMP-9 mRNA—undetectable in controls, detected at 24 h in TB-infected cells and increased at 48 h TIMP-1 mRNA—detected in controls. Only moderate increase at 48 h in infected cells *Human data* (Represented as activity on zymogram and as MMP/CSF-leukocyte ratio): MMP-9 activity in TBM > other meningitides MMP/CSF leukocyte ratio in TBM > other meningitides MMP-9/CSF leukocyte ratio positively associated with neurological complications MMP-2 was constitutionally expressed in the CSF, not affected by infection TIMP-1 was not significantly elevated compared to other meningitides or controls
Thwaites et al. [45]	CNS	Adults (*n* = 21 TBM)	CSF Serum	ELISA	MMP-9 TIMP-1 Also several cytokines	Measured pre- and post-treatment analyte concentrations: All patients received streptomycin (20 mg/kg intramuscularly daily; maximum, 1 g) and an oral regimen of 5 mg/kg isoniazid, 10 mg/kg rifampicin, and 30 mg/ kg pyrazinamide for 3 months, followed by 3 drugs (isoniazid, rifampicin, and pyrazinamide) for 6 months Pre-treatment: MMP-9 = 146 ng/mL, TIMP-1 = 463 ng/mL Post-treatment: MMP-9 = 70 ng/mL (*p* < 0.05), TIMP-1 = 269 ng/mL (*p* > 0.05) MMP-9 was not associated with outcome post-treatment was not significantly different to pre-treatment concentrations
Lee et al. [46]	CNS	Adults (*n* = 24 TBM, 23 acute aseptic meningitis, 10 controls [4 pTB and 6 non-inflammatory neurological disorders])	CSF	ELISA Gelatin zymography	MMP-9 MMP-2	Measured MMP concentrations early (<7 days after treatment) and late (after 7 days of treatment—range 10–106 days) MMP-9: early = 74 ng/mL, late = 123 ng/mL Both early and late TBM concentrations > aseptic meningitis and controls (*p* < 0.001) MMP-2: early = 75 ng/mL, late = 120 ng/mL Early TBM > controls (*p* < 0.01) and late TBM > aseptic meningitis (*p* = 0.01) and controls (*p* < 0.001) Both MMP-9 and -2 appear to increase temporally (after treatment), but not evaluated statistically MMP-9 and -2 significantly higher in patients with delayed neurological complications (*p* < 0.001 and *p* < 0.01 respectively) MMP-9 correlated with CSF protein and white cell count
Green et al. [47]	CNS	Adults (*n* = 37 TBM)	CSF	ELISA	MMP-1 MMP-2 MMP-3 MMP-7 MMP-8 MMP-9 MMP-10 TIMP-1 TIMP-2 TIMP-4	Study compared the effect of dexamethasone on analyte concentrations relative to a placebo group. Concentrations were measured pre-treatment, on day 5 (3–8), day 30, 60, and 270 Significant decrease in MMP-9 early in dexamethasone treatment (day 5, *p* = 0.01)—suggested this as potential mechanism in which steroids improve outcome in TBM No relationship found between early decrease in MMP-9 and outcome Did not find any relationship between pre-treatment MMP or TIMP concentrations and outcome, except: lower MMP-2 associated with hemiparesis (*p* = 0.02) MMP-9 correlated with CSF neutrophil count (*R*^2^ = 0.52, *p* < 0.001)
Rai et al. [48]	CNS	Adults (*n* = 36 HIV negative, 28 HIV positive)	CSF	ELISA	MMP-2 MMP-9	TBM case MMP > Controls for HIV− and HIV+ MMP-2 and -9 concentrations not associated with patient outcome MMP-2 associated with visual impairment MMP-2 (ng/mL) Control: 60.5 (47.1–69.5) HIV− TBM: 77.3 (38.8–221.9) HIV+ TBM: 83.2 (48.8–286.5) HIV− versus HIV+ *p* > 0.05 MMP-9 (pg/mL) Control: 466.8 (432.1–986) HIV− TBM: 4962.2 (200–5966) HIV+ TBM: 3476.2 (491–5999.4) HIV− versus HIV+ *p* > 0.05
Marais et al. [49]	CNS	Adults (*n* = 34 HIV-associated TBM) Sampled longitudinally and stratified into TBM-IRIS vs TBM-non-IRIS	CSF serum	Luminex ELISA	MMP-1 MMP-2 MMP-3 MMP-7 MMP-9 MMP-10 MMP-12 MMP-13 TIMP-1 TIMP-2 Plus a large number of cytokines and chemokines.	CSF was analyzed 3–5 time points in HIV-TBM (*n* = 34) at TBM diagnosis, initiation of ART (day 14), 14 days after ART initiation, at presentation of TBM-IRIS, and 14 days thereafter. 40 mediators in CSF were compared to blood and between patients who developed TBM-IRIS (*n* = 16) vs those who did not (TBM-non-IRIS; *n* = 18) MMP-9 and TIMP-1 were elevated in CSF at TBM diagnosis, while MMP-1, -3, -7, and -10 were significantly higher in blood HIV-TBM patients who subsequently developed TBM-IRIS showed significantly elevated MMP-1, MMP-7, MMP-10, TIMP-1 and TIMP-2 in the baseline CSF, compared to those who did not
Majeed et al. [50]	CNS	C6 glioma cells (in-vitro study) Adults (*n* = 91 TBM cases, 16 controls)	CSF Serum	Zymography Reverse zymography	MMP-9 TIMP-1	*In-vitro study:* Infected C6 glioma cells with *Mtb,* and treated them with MMP9-inhibitor (SB-3CT) and dexamethasone MMP-9: infected cells > uninfected cells Both drugs decreased MMP-9 concentrations *Human data*: measured MMP-9 and TIMP-1 levels at presentation (grouped according to TBM staging) CSF and serum MMP-9: ○ cases > controls for all 3 stages ○ higher MMP-9 concentrations at more severe TBM stages CSF TIMP-1 detectable in controls and stage I TBM only. Serum TIMP undetectable MMP-9 (ng/mL) *CSF* concentrations: Controls: 0.62 ± 0.4, Stage I TBM: 9 ± 0.87, Stage II TBM: 12 ± 1.34, Stage III TBM: 16.9 ± 2.7 MMP-9 (ng/mL) *serum* concentrations: Controls: 5.67 ± 2.45, Stage I TBM: 830.66 ± 83.07, Stage II TBM: 1202.55 ± 136.81, Stage III TBM: 1679 ± 277.4
Marais et al. [51]	CNS	Adults (*n* = 34 as described above in Marais et al. [49])	RNA from blood, Proteins from CSF and plasma	Microarray analysis (RNA) ELISA (CSF and plasma)	>47,000 probes	Whole blood transcriptional signature confirmed elevated MMP-8 and MMP-9, with MMP-9 elevated at protein level in baseline CSF in TBM-IRIS patients MMP-9 transcript remained elevated despite anti-TB and anti-inflammatory (prednisone) treatment in TBM-IRIS patients, possibly leading to degradation of the extracellular matrix as a mechanism of tissue damage in the central nervous system that is systemically reflected in the blood
Mailankody et al. [52]	CNS	Adults (*n* = 40 TBM)	CSF	ELISA	MMP-9 TIMP-1	MMP-9 positively correlated with GCS (*r* = 0.3, *p* = 0.04) and CSF albumin levels (*r* = 0.354, *p* = 0.025) MMP-9 negatively correlated with CSF glucose (*r* = −0.4, *p* = 0.007) MMP-9 did not correlate with Q-alb or CSF neutrophil count Higher median MMP-9 in patients with good outcome (254 ng/mL) compared to bad outcome (192 ng/mL), but not significant
Li et al. [53]	CNS	Children (*n* = 40 TBM, 8 controls)	CSF (lumbar & ventricular) Serum	Luminex	MMP-9 MMP-2 TIMP-1 TIMP-2	First study to measure MMP (and TIMP) concentrations in children with TBM Persistently elevated gelatinase (and inhibitor) concentrations in paediatric TBM MMP-9 levels decreased early in response to TB treatment (including steroid usage), but increased later during the treatment phase No correlation between admission gelatinase levels and Q-alb, clinical characteristics or poor outcomes Better outcomes (risk ratio 2.1, 95% CI 1.23–3.53, *p* = 0.01) at 6 months for patients whose MMP-9 levels increased during sampling period

Data presented as mean ± standard deviation or median (range) depending on information available from the study. * Provide diagnostic accuracy analysis. Matrix metalloproteinase (MMP), tissue inhibitor of matrix metalloproteinase (TIMP), Enzyme-linked immunosorbent assay (ELISA), broncho-alveolar lavage (BAL), interleukin (IL), cerebrospinal fluid (CSF). Anti-retroviral therapy (ART). Area under the curve (AUC). Cross-linked C-telopeptide of type I (CTX-I) and type III (CTX-III) collagen. Extracellular matrix metalloproteinase inducer (EMMPRIN). Procollagen I N-terminal propeptide (PINP). Procollagen III N-terminal propeptide (PIIINP). Procollagen III C-terminal propeptide (PIIICP). Platelet-derived growth factor-BB (PDGF-BB). Platelet factor-4 (PF4). Pentraxin-3 (PTX3). Receiver operating characteristics (ROC). Tumour necrosis factor (TNF). Regulated upon Activation, Normal T cell Expressed, and Secreted (RANTES). Membrane type (MT). Monocyte chemoattractant protein 1 (MCP-1). Haematoxylin and Eosin (H&E). Single nucleotide polymorphism (SNP). Albumin quotient (Q-alb).

**Table 2 ijms-20-01350-t002:** Evaluation of MMP inhibitors in pulmonary TB animal models.

MMP Inhibitor	Animal Model	Treatment Started *	Combination Therapy	Results in Treated Group Compared to Controls				Reference
				Lung CFU	Lung Pathology	Mortality	Other Findings in Treated Group	
Batimastat (BB-94)	Mouse (Balb/c)	Day 1	No	NR	+	+	Lower TNF-α, IL-2 and IL-1α. Higher IL-4	[113]
		Day 30	No	NR	=	=	No difference in TNF-α, IL-2 and IL-4. Higher IL-1α in pneumonic areas.	[113]
	Mouse (C57BL/6)	Day 18	No	−	−/= ^a^	NR	Decreased leukocytes. No differences in IFN-γ, IL-4, IL-12, TNF-α and IL-10. Less CFU in blood.	[114]
		Day 1	No	=	NR	NR	Less CFU in spleen and blood by day 14.	[115]
		Day 7	Isoniazid	−	NR	NR		[118]
Cipemastat (Ro 32-3555, Trocade)	Mouse (C3HeB/FeJ)	Day 1	No	=	+	+	Higher rate of cavitation	[119]
	Rabbit (New Zealand white)	Day 35	No	NR	+	NR	Higher rate of cavitation. No differences in disease severity by gross pathology or histology	[116]
Marimastat (BB-2516)	Mouse (C57BL/6)	Day 7	No	=	=	NR		[118]
		Day 7	Isoniazid	−	−	NR	Improved stability of blood vessels surrounding TB lesions	[118]
Prinomastat	Mouse (C57BL/6)	Day 7	Isoniazid	=	NR	NR		[118]
SB-3CT	Mouse (C57BL/6)	Day 7	Isoniazid	−	NR	NR		[118]
MMP-9 inhibitor I	Mouse (C57BL/6)	Day 7	Isoniazid	−	NR	NR		[118]
Anti MMP-9 antibody	Mouse (C3HeB/FeJ)	Day 42	Rifampin, isoniazid and pyrazinamide	−	−	NR	Less relapse rates after 12 weeks of treatment	[117]
CC-3052 (PDE-4 inhibitor) ^b^	Rabbit (New Zealand white)	Day 28	No	=	+	NR	Worse disease compared to untreated controls	[111]
		Day 28	Isoniazid	−	−	NR	Less inflammation and fibrosis compared to isoniazid monotherapy controls	[111]
Doxycycline	Guinea pig	Day 14	No	−	−	NR	No MMP-specific effect of doxycycline was identified	[28]

* Days post-infection with *M. tuberculosis*, (−) equals less disease, (+) equals worse disease, (=) no change. NR = not reported, ^a^ Smaller pulmonary granulomas with more collagen deposition by day 40. No difference in pathology compared to controls by day 60. ^b^ Phosphodiesterase-4 inhibitor. Tumor necrosis factor (TNF), interleukin (IL), colony forming units (CFU).

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
