# Peer review of "Matrix Metalloproteinases in Pulmonary and Central Nervous System Tuberculosis—A Review"

_ijms, 2019, doi:10.3390/ijms20061350_

Round 1
Reviewer 1 Report
The manuscript entitled “Matrix metalloproteinases in pulmonary and central nervous system tuberculosis – a review” describes the potential role of MMPs in pulmonary and CNS TB, provides the different methods of MMP investigation and discusses the translational implications of MMP inhibition to reduce immunopathology. The contents, figures and tables provided by authors in this review let reader easy comprehension.Author Response
We thank the reviewer for these positive comments.
Reviewer 2 Report
This review article entitled “Matrix metalloproteinases in pulmonary and central
3 nervous system tuberculosis” by Rohlwink et al. is a thorough overview of recent studies on roles of MMPs in pulmonary TB and TBM. TB remains a serious disease causing many deaths globally, and with the antiviral treatment against HIV, causing immune reconstitution inflammatory syndrome, TB adds a new concern to public health. Therefore, the topic of this article has a high clinical impact. The article starts with a well-organized introductory section for cellular biology of MMPs, and it documented the involvement of different types of MMPs in different aspects of pTB in a very compelling manner. As for TBM, the studies are rather contradictory, which reflects the fact that the field remains to be further developed. This article is expected to attract wide range of readership. There are only a few minor points to be clarified.
Figure 1 legend does not mention every item illustrated in the figure. In fact, none of the key words such as “Granuloma”, “Cavity”, “Necrosis”, “Transmission” appears. Although it is clearly indicated in the main text, legend itself should explain what the figure indicates.
In line207, it is described that Mtb increases TIMP1 upon binding to collagen, and lots of studies listed in Table 1 seem to agree with this result. However, TIMP1 is indicated to be decreased in Fig. 1. Which study is this statement based on? It would be helpful to include citations in the figure.
Author Response
We thank the reviewer for the comments and for pointing out the incompleteness of the legend for Figure 1. We have expanded the legend to be more explanatory and to incorporate all keywords. We opted not to include citations in the figure in order to avoid it becoming very cluttered.
Regarding TIMP1 in the text versus the figure, we have added an additional sentence to clarify the complexity of current knowledge based on the literature: "Whilst plasma concentrations of TIMP-1 are elevated in pulmonary TB in adults and children (Chen at al, Mol Med Rep 2017; Pavan Kumar at al, Clin Vaccine Immunol 2013), the concentration of antiproteases at the site of disease is evidently insufficient to prevent matrix destruction, and TIMP-1 concentrations in pulmonary secretions are reduced in TB (Elkington et al, J Clin Invest 2011). In cell culture experiments, the increase in MMP secretion is not countered by an increase in TIMP-1 secretion (Elkington et al, J Immunol 20015), consistent with Mtb skewing the protease-antiprotease balance towards matrix breakdown." We hope the combination of the above clarification to the manuscript text and the expanded figure legend have answered the queries raised by the reviewer.